# Crystallinity Dependence of PLLA Hydrophilic Modification during Alkali Hydrolysis

**DOI:** 10.3390/polym15010075

**Published:** 2022-12-25

**Authors:** Jiahui Shi, Jiachen Zhang, Yan Zhang, Liang Zhang, Yong-Biao Yang, Ofer Manor, Jichun You

**Affiliations:** 1Key Laboratory of Organosilicon Chemistry and Material Technology, Ministry of Education, College of Material, Chemistry and Chemical Engineering, Hangzhou Normal University, Hangzhou 311121, China; 2School of Chemistry and Chemical Engineering, Liaoning Normal University, Dalian 116029, China; 3The Wolfson Department of Chemical Engineering, Technion-Israel Institute of Technology, Haifa 32000, Israel

**Keywords:** PLLA, hydrophilicity, alkali hydrolysis, crystallinity

## Abstract

Poly(L-lactic acid) (PLLA) has been extensively used in tissue engineering, in which its surface hydrophilicity plays an important role. In this work, an efficient and green strategy has been developed to tailor surface hydrophilicity via alkali hydrolysis. On one hand, the ester bond in PLLA has been cleaved and generates carboxyl and hydroxyl groups, both of which are beneficial to the improvement of hydrophilicity. On the other hand, the degradation of PLLA increases the roughness on the film surface. The resultant surface wettability of PLLA exhibits crucial dependence on its crystallinity. In the specimen with high crystallinity, the local enrichment of terminal carboxyl and hydroxyl groups in amorphous regions accelerates the degradation of ester group, producing more hydrophilic groups and slit valleys on film surface. The enhanced contact between PLLA and water in aqueous solution (i.e., the Wenzel state) contributes to the synergistic effect between generated hydrophilic groups and surface roughness, facilitating further degradation. Consequently, the hydrophilicity has been improved significantly in the high crystalline case. On the contrary, the competition effect between them leads to the failure of this strategy in the case of low crystallinity.

## 1. Introduction 

With the increasing requirement of health care and the improved understanding of natural tissue, more and more attention is being paid to medical techniques. Tissue engineering, as a newly emerged field, aims at the development of functional substitutes for damaged tissue [1]. In tissue engineering, one of the ideal characteristics for the designed and manufactured medical material is that its surface can lead to increased affinity of biomolecules [2]. Poly(lactic acid) (or polylactide, abbreviated as PLA) is a semi-crystalline and environmentally friendly material used in tissue engineering in recent decades. Lactic acid can be produced by an industrial process from petrochemical feedstocks, or alternatively by a fermentation process from renewable sources such as corn, wheat or rice. The difference is that the petrochemical route results in the racemic mixture of D- and L-enantiomers, while the production from the natural route is mostly (99.5%) the L-isomer which leads to poly(L-lactic acid) (PLLA) in the next step [3,4]. 

Although PLLA has excellent biocompatibility and biodegradability properties and thus has been widely used in medical purposes such as implants, bone grafting and fracture fixation devices [5,6], it is in nature a relatively hydrophobic material with a static water contact angle in the range of 75°–85° [7,8,9]. For biomedical applications such hydrophobicity will lead to low cell affinity and thus affect the cell adhesion onto its surface. In order to improve the cytocompatibility of PLLA, many surface modification techniques have been proposed, including polymerization grafting [10], ozone oxidization [11,12], plasma treatment [13,14,15,16], alkaline hydrolysis [17,18], surface coating [19], layer-by-layer self-assembly [20,21], etc. 

In addition to relatively high cost, low efficiency or residual toxic solvent, the reported strategies always concern either complicated chemical reaction or special requirement of equipment, which does limit their applications [22,23]. The hydrolysis of PLA as a potential method has attracted the attention of researchers. Previous results pay more attention to the evolution of PLA’s matrix properties [24,25,26] rather than its surface properties. 

The degradation of ester bonds with the help of alkaline hydrolysis treatment has been considered as a reliable, green, low labor-intensive and less time-consuming method [27,28,29]. It is well known that in alkaline hydrolysis ester bonds undergo the nucleophilic attack of hydroxide ions, and finally the ester bonds are cleaved to create carboxyl and hydroxyl groups. By varying the chemical composition of material surface, alkaline hydrolysis can improve PLA’s hydrophilicity and reduce its water contact angle. Consequently, much effort has been made to develop the method of alkaline hydrolysis for PLLA in recent years. For instance, using ethanol to assist the hydroxide nucleophilic attack on ester bond in NaOH treatment, Yang et al. modified the surface of the microporous PLLA film to enhance its cell affinity [17]. Sabee and his co-workers investigated the dependence of surface modification for PLLA microspheres on sodium hydroxide (NaOH) concentration [30]. For the preparation of bioactivated PLLA scaffolds, the flexible surface functionalization by the specific alkaline treatment has been successfully developed by Meng et al. [18]. Tsuji et al. investigated the stability and degradation of PLLA in NaOH. Their results indicated that the hydrolysis took place preferentially in the amorphous regions. The specimen with low crystallinity, therefore, exhibited shorter degradation period. 

In this work, therefore, an alkaline hydrolysis strategy, inspired by the preferential degradation in amorphous region, is proposed to establish a more convenient, effective and green approach for the surface modification of PLLA with the consideration of its crystallinity. The crystallinity dependence of PLLA hydrophilic modification during alkali hydrolysis has been investigated in detail. The random degradation on the specimen with low crystallinity and selective degradation on alternating structures including crystal lamellar and amorphous region are compared. Based on the surface chemical composition (obtained from XPS and FTIR) and surface roughness (determined by SEM and AFM), the important role of crystallinity in alkali hydrolysis has been clarified. 

## 2. Experiment

### 2.1. Materials

PLLA (3001D, *M*w = 120,000, PDI = 1.9) was purchased from Nature works, USA. Sodium hydroxide (NaOH, AR, ≥96.0%) was bought from Lingfeng Chemical Reagent Co., Ltd., Shanghai, China. 

### 2.2. Preparation and Hydrolysis of PLLA Films

PLLA films were prepared by hot-pressing at 200 °C and 10 MPa (with a thickness of 500 μm). The melted PLLA films were quickly cooled to 120 °C under 10 MPa and annealed for 60 min, and were named as PLLA_60_. The unannealed films were named as PLLA_0_. After heat treatment procedure, all PLLA films were quenched in an ice-water mixture at 0 °C to stop crystallization. Hydrophilic modification of PLLA film was done as follows by surface alkaline hydrolysis. Firstly, the PLLA films were immersed in 1 mol/L NaOH solution, and alkaline hydrolysis was performed at 55 °C for a predetermined time (1 min, 2 min, 3 min, 4 min, 5 min, 10 min, 15 min, 20 min and 25 min). Secondly, the treated PLLA films were rinsed with deionized water and soaked for 2 h. Finally, the films were dried in a vacuum oven at 55 °C for 48 h.

### 2.3. Characterization 

Differential scanning calorimetry (DSC Q2000, TA) measurements were performed in a nitrogen atmosphere at the heating and cooling rate of 10 °C/min. A universal tensile testing machine (Instron 5966) was used for tensile property tests at room temperature. The tensile speed was 10 mm/min. The PLLA samples (before or after alkaline hydrolysis) were cut into dumbbell shapes (18 × 0.5 × 3 mm, ASTM D 412-80). At least five specimens were tested for each sample, and the result of the tensile property test was the average value of 5 stretches. The surface structure of the PLLA samples was observed by scanning electron microscopy (SEM, Hitachi S-4800) and atomic force microscopy (AFM, Seiko E-Sweep). AFM was used to measure the surface roughness of PLLA films, in which a tip with a force constant of 2 N/m was adopted. The water contact angle of the PLLA films was obtained by drop-shape analysis (DSA, Krüss DSA100). The surface composition of the PLLA film was characterized by means of Thermo ESCALAB 250 X-ray photoelectron spectrometer (XPS) and Fourier transform infrared resonance (FTIR, Brucker VERTEX 70 V) with attenuated total reflection (ATR) mode.

## 3. Results and Discussion

### 3.1. DSC Analysis of the Crystallinities of PLLA Films

First of all, the crystallization kinetics of PLLA at 120 °C (isothermal crystallization) were investigated with the help of DSC. As shown in Figure 1A, there is an obvious exothermic peak at the beginning stage, corresponding to the crystallization of PLLA. It takes ~41 min for PLLA to fully crystallize. In this work, two specimens were adopted to clarify the crystallinity dependence of alkali hydrolysis. They were named as PLLA_0_ and PLLA_60_ since these specimens were crystallized at 120 °C for 0 min and 60 min, respectively. Then DSC measurements were performed to identify the crystallinities of them. In the black curve in Figure 1B, the glass transition temperature is located at 62.3 °C. During heating, there is a broad exothermic peak. It starts at ~90 °C and does not disappear until the occurrence of endothermic peak. The endothermic peak can be attributed to the melting of PLLA. On the contrary, the exothermic peak arises from the cold crystallization of PLLA since the temperature is between its glass transition temperature and melting temperature. It is facile to calculate the crystallinities of PLLA according to Equation (1).
(1)$$X_{w}=\frac{\Delta H_{m}}{\Delta H_{m}^{0}}$$
where *X_w_*, $\Delta H_{m}$ and $\Delta H_{m}^{0}$ are crystallinity, the measured melting enthalpy and standard melting enthalpy (93.7 J/g [31]), respectively. For PLLA_0_, its crystallinity is only 4.1%. In the red curve, PLLA_60_ has a similar glass transition temperature to PLLA_0_. Its melting temperature, however, exhibits a slightly higher magnitude (166.2 °C) relative to PLLA_0_ (165.4 °C). Such an improvement of melting temperature can be attributed to the thicker crystal lamellae resulting from sufficient isothermal crystallization. As for the calculated crystallinity of PLLA_60_, it is 46.3%. Because of the high crystallinity in PLLA_60_ during isothermal crystallization, no cold crystallization (exothermic peak) is observed in the red curve. 

### 3.2. DSA and Tensile Test during Alkali Hydrolysis

Figure 2A gives the hydrolysis time dependence of water contact angle of PLLA films with different crystallinities based on the DSA method. In agreement with previous reports [7,8,9], the water contact angles of PLLA_0_ and PLLA_60_ films before alkali hydrolysis are located at ~74.0°, suggesting that they are not very hydrophilic materials. Upon alkali hydrolysis, the evolution of contact angle exhibits remarkable dependence on the crystallinities of PLLA. The pictures showing hydrophilicity through contact angle measurement have been provided in Appendix A. For PLLA_0_ films (black line in Figure 2A), the contact angle did not change significantly within the experiment period (25 min) but only increased slightly. For the annealed PLLA films (PLLA_60_), there were two stages in the evolution of the water contact angle. At the first one, the contact angle decreased quickly with alkali hydrolysis time (down to 22.4° at 15 min), while at the second stage (15 min to 25 min), the value was not much sensitive to alkali hydrolysis anymore. In other words, the surfaces of PLLA films with higher crystallinity become hydrophilic if the hydrolysis time is long enough. A facile strategy, therefore, has been developed to tailor the surface wettability of PLLA via alkali hydrolysis. In the assessment of mechanical performance, our attention will be paid only to the specimen of PLLA_60_ since for PLLA_0_ its wettability is hard to manipulate by means of hydrolysis. As shown in Figure 2B, the elongation at break of untreated PLLA_60_ film is ~4.5% and the yield strength is ~60 MPa. After alkali hydrolysis, those two values decrease to 3.1% and 55 MPa, respectively. Both variations are within the experimental errors. This result indicates that the alkali hydrolysis takes place only on the film surface and has no remarkable influence on their mechanical properties in the period up to 25 min. This is significant for the modified PLLA in medical applications. 

In some special scenarios (e.g., fog capture), hydrophilicity or hydrophobicity gradient play key roles. According to the strategy developed as discussed for Figure 2A, it is facile to fabricate a wettability gradient with the help of home-made setup (Figure 3A). During the experiment, a specimen (PLLA_60_) with certain length (2.5 cm, here) was kept in alkali aqueous solution and drawn out with the speed of 1 mm/min. The top part of the specimen corresponded to a shorter (down to 0 min) alkali hydrolysis time, contributing to higher magnitudes of water contact angles (Figure 3C). This can also be verified by the hemispherical droplets at the left part in Figure 3B. On the contrary, the bottom part stayed in alkali aqueous solution for a longer period (up to 25 min), yielding complete wetting of water on the specimen upon sufficient alkali hydrolysis (right parts of Figure 3B,C). Based on the results discussed above, the manipulation of the wettability gradient on PLLA films has been achieved successfully by varying the alkali hydrolysis period. The tunable water contact angle can meet unique requirements for the customized surface in various applications. Obviously, the adjustment of water contact angles (Figure 2A) and its wettability gradient (Figure 3B,C) are under the control of both surface composition and structures [16,21]. In the following parts, we will discuss them one by one. 

### 3.3. XPS and FTIR Characterization during Alkali Hydrolysis

When the specimen is kept in alkali aqueous solution, the degradation of PLLA takes place, producing carboxyl and hydroxyl groups, which is shown in Equation (2) [32].
R-COO-R′ + NaOH → RCOO^−^ + HO-R′ + Na^+^(2)

During alkali hydrolysis, the surface composition was identified with the help of XPS and FTIR in ATR mode. Figure 4A shows the C1s spectra of specimens before and after alkali hydrolysis. There are three characteristic peaks at 284.6 eV (Peak 1), 286.8 eV (Peak 2) and 288.8 eV (Peak 3), corresponding to C-C, C-O-C and C=O groups, respectively [33,34]. In the black curve, Peak 2 exhibits higher area fraction relative to Peak 3. As for hydrolysis on the specimen of PLLA_0_, they are comparable (red curve). While in the case of PLLA_60_ after treatment, Peak 3 corresponds to higher area fraction (blue curve) compared with Peak 2. The lower magnitude of area ratio between Peak 2 and Peak 3 indicates that the hydrolysis of PLLA takes place, during which the ester bond has been broken and converted into hydroxyl and carboxyl groups (as Equation (2)). In this process, the accounts of C-O-C (Peak 2) in XPS C1s spectra decrease. This conclusion has also been supported by FTIR results as shown in Figure 4B. To compare the absorbance of specimen before and after hydrolysis, all curves have been normalized according to the characteristic peak of -CH_3_ at 1360 cm^−1^. The characteristic peak at 1180 cm^−1^ is related to the stretching vibration of C-O-C bond in PLLA. It exhibits a higher intensity before hydrolysis (black curve). The treatment for PLLA_0_ leads to the reduced absorbance of the corresponding peak (red curve). Such decrease has been further enhanced in the case of PLLA_60_ upon hydrolysis (blue curve). The XPS (Figure 4A) and FTIR (Figure 4B) results reveal that some ester bonds have been cleaved during alkali hydrolysis. In Figure 4C and 4D, the peaks at 3647.3 cm^−1^ and 1717.9 cm^−1^ can be indexed to the hydroxyl and carboxyl groups, respectively. The absorbances of them at the indicated position exhibit much higher magnitudes relative to the specimen before treatment, suggesting the higher content of carboxyl and hydroxyl groups after alkali treatment. The combination of degradation of ester group and generation of carboxyl/hydroxyl groups makes it clear that alkali hydrolysis cleaves the ester bond and produces hydrophilic groups. Furthermore, the comparison between red and blue curves in Figure 4 reveals that there are more hydrophilic (carboxyl and hydroxyl) groups in the specimen of PLLA_60_.

### 3.4. SEM and AFM Characterization during Alkali Hydrolysis

As mentioned in the Introduction, the surface roughness of the PLLA films may be varied during alkaline hydrolysis. In this work, SEM has been employed to characterize the surface morphology of PLLA films. From Figure 5A to Figure 5C, it can be seen that the alkaline hydrolysis happens randomly on the surface of PLLA_0_ films, producing concave structures (marked by red arrows). As the time of alkaline hydrolysis increase, the density and diameter of the concave structures gradually increase. Finally, they connect with each other, contributing to higher magnitudes of surface roughness. In the case of PLLA_60_, the resultant morphologies are of great difference, as shown in Figure 5D–F. There are some slit valleys in Figure 5D (hydrolysis for 1 min). Then their size increases while some new valleys appear in Figure 5E (hydrolysis for 5 min). Finally, these structures crowd together, accounting for the significant improvement of surface roughness. The reason for the occurrence of these slit valleys (marked by green arrows) will be discussed in the following parts. The SEM results in Figure 5 make the following points clear. On one hand, the alkali hydrolysis on PLLA_0_ and PLLA_60_ produces concave and slit structures, respectively; on the other hand, the specimen of PLLA_60_ exhibits rougher surface relative to PLLA_0_. 

To identify the surface roughness upon hydrolysis quantitatively, AFM has been employed. The 3D AFM images are shown in Figure 6. Obviously, the surface becomes more and more rough during the alkali treatment in PLLA_0_ (Figure 6A–C). The increase of surface roughness in PLLA_60_ (Figure 6D–F) is more remarkable. Therefore, the AFM topography images have excellent agreement with SEM images. The root–mean–square (RMS) surface roughness has been calculated with the help of commercial software equipped on AFM based on Equation (3).
(3)$$\mathrm{Roughness}=\sqrt{\frac{1}{S}\iint {\left\{F\left(x,\;y\right)-Z\right\}}^{2}dxdy}$$
where *S* and *Z* are the corresponding area and average height at the calculated position, respectively. The resultant surface roughness is shown in Figure 6G. At the surface of hot-pressed film, the RMS roughness of PLLA_0_ and PLLA_60_ is located at ~100 nm, suggesting a relatively flat film surface. After the treatment in alkali aqueous solution, the roughness exhibits higher magnitudes. In the case of PLLA_0_, it increases to 156 nm (5 min), 279 nm (15 min) and 395 nm (25 min). The alkali hydrolysis on PLLA_60_ results in a more significant increase in RMS roughness. It is 178 nm and 406 nm upon treatment for 5 min and 15 min, respectively. Finally, it reaches 537 nm at 25 min. Both AFM images and corresponding surface roughness are all consistent with SEM results shown in Figure 5. According to the discussion above, the alkali hydrolysis destroys the film surface, contributing to the rough surface and playing an important role in the consequent wettability. This treatment produces a much rougher surface in the specimen with higher crystallinity (PLLA_60_).

### 3.5. PLLA Surface Alkali Hydrolysis Mechanism

According to the discussion above, we can describe the hydrophilicity of PLLA in NaOH solution as well as its crystallinity/hydrolysis time dependences as follows. When the specimen with high crystallinity (PLLA_60_) is kept in alkali aqueous solution, the degradation of PLLA takes place. As a result, the ester bond has been cleaved, producing carboxyl and hydroxyl groups. Both are beneficial to the improvement of PLLA hydrophilicity. At the same time, the degradation of PLLA yields rough surface. The roughness can enhance the hydrophilicity further, accounting for water contact angles down to 20° (Figure 2A and Figure 3). However, the reason that such a hydrophilicity evolution cannot happen to the amorphous PLLA films remains obscure. It is well known that surface roughness and chemical composition are both important factors affecting the hydrophilicity of PLLA. Our attention, therefore, has been paid to the following points to understand the difference between PLLA_0_ and PLLA_60_. 

(1) The densities of hydrophilic groups, defined as the number of these groups per unit area, are of great difference in specimens with low and high crystallinities. In PLLA film with very low crystallinity (PLLA_0_), there are only terminal carboxyl and hydroxyl groups in PLLA chain. They randomly distribute on the whole surface (Figure 7A), leading to the lower density of the hydrophilic groups. However, for a PLLA film upon sufficient crystallization, the terminal carboxyl and hydroxyl groups have been expelled out from crystal lamellae and locate in the inter-lamellar regions (Figure 7B). The enrichment of carboxyl and hydroxyl groups contributes to a higher local density of the hydrophilic group. 

(2) When our specimen is immersed in alkali aqueous solution, the local density of the hydrophilic group produces significant influence on the hydrolysis. In PLLA_0_, the surface is not hydrophilic enough to enhance the contact between PLLA film and water since the water contact angle of untreated PLLA locates at 74.0° (Figure 2A). In the case of PLLA_60_, the alkali solution prefers to wet the hydrophilic regions between crystal lamellae, accelerating the hydrolysis. Consequently, more carboxyl and hydroxyl groups have been generated, corresponding to the lower content of C-O-C in XPS results (Figure 4A) and enhanced absorbance in FTIR (Figure 4C, D). 

(3) The surface roughness resulting from hydrolysis exhibits crucial dependence on the crystallinity. In the case of PLLA_0_, the hydrolysis takes place randomly at the whole film surface, contributing to concave structures. The density of hydrophilic groups is not high enough to induce the sufficient contact between PLLA and water. The competition effect between chemical composition and roughness is the reason for the Cassie state in which there are air cushions, depressing the further degradation. The situation in PLLA_60_, however, is different. The area containing PLLA crystal lamellae remains stable while the hydrophilic area in inter-lamellar regions accelerates the hydrolysis since it occurs preferentially in the amorphous region of PLLA films [32]. The selective hydrolysis results in slit valleys shown in SEM (Figure 5) and AFM (Figure 6) images, and higher magnitudes of height difference in two regions, yielding higher surface roughness (Figure 6G). There are so many hydrophilic groups in the valleys that the contact between PLLA and water has been enhanced, accounting for the Wenzel state in which sufficient contact can promote further hydrolysis. This is the so-called synergistic effect. 

(4) With the combination of surface roughness and chemical composition, the specimen of PLLA_60_ exhibits strong hydrophilicity and lower water contact angle (down to 20°). In the case of PLLA_0_, the hydrolysis has been slowed down, leading to lower content of carboxyl and hydroxyl groups on film surface. It is difficult to enlarge the hydrophilicity by surface roughness. The water contact angle, therefore, increases slightly upon hydrolysis. In one word, the competitive effect and synergistic effect between surface roughness and chemical composition dominate the wettability in PLLA_0_ and PLLA_60_, respectively. 

## 4. Conclusions

In this work, a green and convenient strategy of surface modification for PLLA film has been developed via alkaline hydrolysis during which the crystallinity of PLLA plays an important role. In the case of high crystallinity, the water contact angle decreases remarkably upon alkali hydrolysis (from 74.0° down to 18.2°). The evolutions of surface roughness (from ~100 nm up to 537 nm) and composition indicate that the local enrichment of terminal carboxyl and hydroxyl groups during isothermal crystallization contributes to sufficient contact between PLLA and water in aqueous solution, accelerating the degradation of PLLA. This is the reason for high-density carboxyl and hydroxyl groups and higher magnitudes of surface roughness in the high crystalline case. The synergistic effect between them promotes the further alkaline hydrolysis and improves the hydrophilicity of PLLA. On the contrary, terminal hydrophilic groups distribute on the whole surface randomly in the specimen with low crystallinity, which leads to limited hydrophilic modification. In that case, the surface of PLLA is still hydrophobic and the increase of surface roughness contributes to the Cassie state in which there are air cushions, depressing the further degradation. As a result, the competition effect between surface roughness and composition leads to failure of this strategy. Note that in this work, our study was performed at fixed temperature and pH levels. The conditions of extreme temperature and pH may result in a complicated Cassie state, a Wenzel state and even transition between them which will be investigated in future publication. Our results are significant for the surface modification of PLLA as well as its applications in medical fields. 

## Figures and Tables

**Figure 1 polymers-15-00075-f001:**
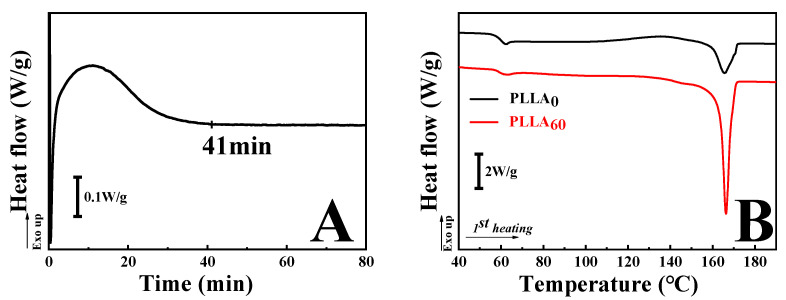
The crystallization kinetics of PLLA film at 120 °C (**A**) and the first heating DSC curve of PLLA_0_ and PLLA_60_ (**B**).

**Figure 2 polymers-15-00075-f002:**
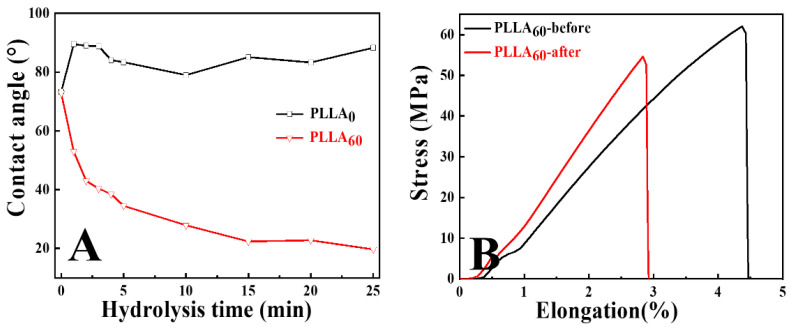
The hydrolysis time dependence of water contact angle in PLLA_0_ and PLLA_60_ (**A**) and strain–stress curves of PLLA_60_ before and after hydrolysis (**B**).

**Figure 3 polymers-15-00075-f003:**
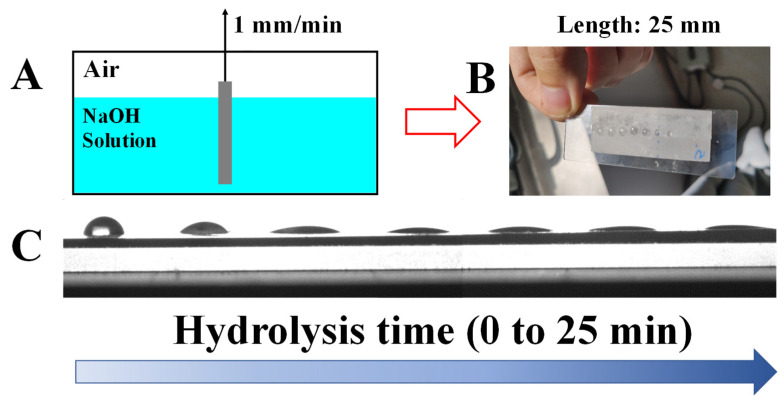
The home-made setup for the fabrication of hydrophilic gradient via alkali hydrolysis (**A**). (**B**,**C**) show the photos of PLLA films with tunable water contact angle.

**Figure 4 polymers-15-00075-f004:**
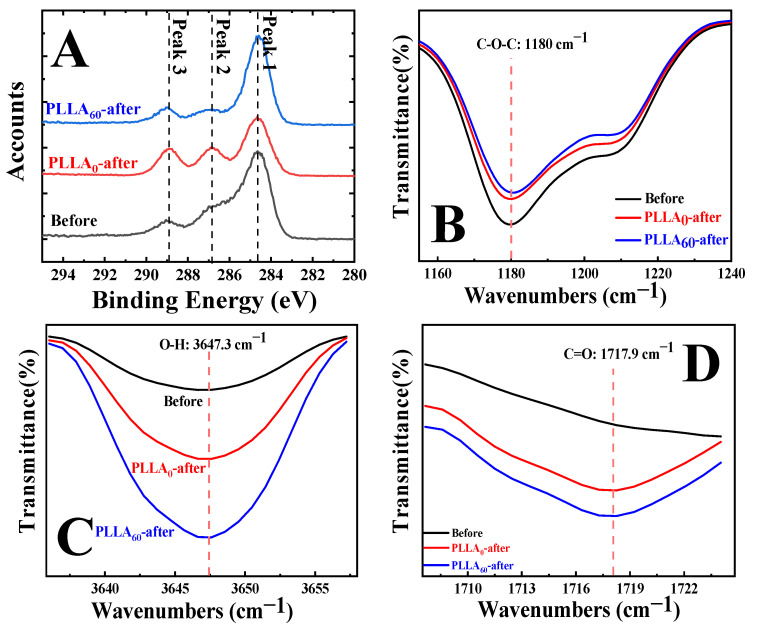
XPS C1s spectra (**A**) and FTIR spectra (**B**–**D**) of specimens before and after alkali hydrolysis of PLLA_0_ and PLLA_60_.

**Figure 5 polymers-15-00075-f005:**
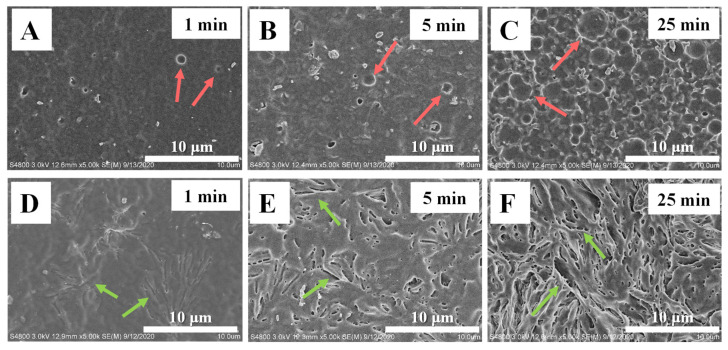
SEM images of modified PLLA membranes upon alkali hydrolysis for the indicated time in PLLA_0_ (**A**–**C**) and PLLA_60_ (**D**–**F**).

**Figure 6 polymers-15-00075-f006:**
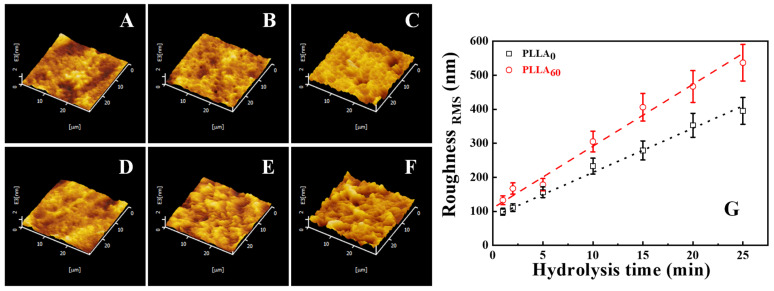
AFM images of modified PLLA membranes: PLLA_0_ with alkaline hydrolysis time for 1 min, 5 min and 25 min (**A**–**C**); PLLA_60_ with alkaline hydrolysis time for 1 min, 5 min and 25 min (**D**–**F**); (**G**) shows the roughness evolution during hydrolysis in PLLA_0_ and PLLA_60_.

**Figure 7 polymers-15-00075-f007:**
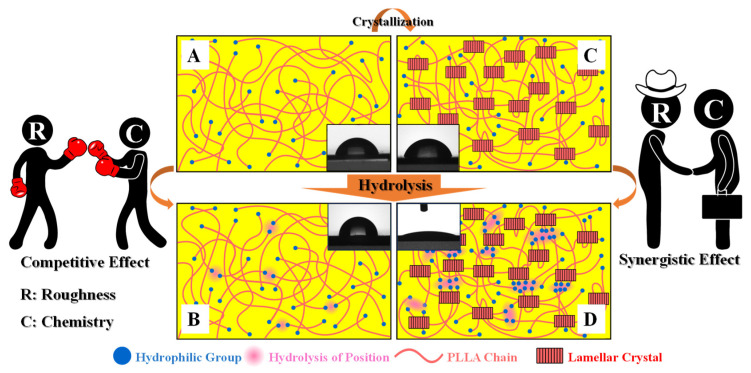
Schematic diagram of PLLA surface alkali hydrolysis mechanism: (**A**) Untreated amorphous PLLA film surface; (**B**) amorphous PLLA film surface after hydrolysis; (**C**) untreated crystallized PLLA film surface; (**D**) crystallized PLLA film surface after hydrolysis.

## Data Availability

The data presented in this study are available on request from the corresponding author.

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
