# Peer review of "Crystallinity Dependence of PLLA Hydrophilic Modification during Alkali Hydrolysis"

_polymers, 2022, doi:10.3390/polym15010075_

Round 1

Reviewer 1 Report

In this manuscript the process of alkali hydrolysis has been used to produce hydrophilic surface through green strategy. PLLA is successfully cleaved in respective carboxyl and hydroxyl groups to facilitate hydrophilic surface, however roughness of the surface resulted upon degradation of PLLA and upon low crystallinity this strategy is no more valid. and overall the manuscript is not properly formatted, headings with specific characterization techniques are missing.

Although an interesting research, however many improvements might be done in the final version as suggested below. 1. First of all the line no in the entire manuscript is missing, formatting is not proper 2. In the experimental , 2.1 materials section , please mention the percentage purity of NaOH. 3. In the result and discussion section, please specify each result with the segregated headings for characterization. 4. In figure 1 a and b please mention the values on y-axis. 5. Figure 2 A and B showing contact angle measurement, or hydrophilicity please specify the characterization technique that is discussing under specific heading, and please provide pics showing hydrophilicity through contact angle measurement. 6. Figure 3 should be improved or make it more practically to show clarity. 7. In Figure 4 b, c and d, please specify the respective hydroxyl and carboxyl groups in FTIR peaks 8. Figure 5 resolution should be improve, and please mark the arrows or circles to clearly show the surface roughness. 9. In figure 6 the image is not clear and axis value are not readable, please improve the resolution. 10. Figure 7 must be added after abstract or can be added as graphical abstract. please improve the resolution, some words are not readable. 11. In the conclusion section, please mention how much decrease in contact angle measurement,  and must add some quantitative data to aid final conclusion. Further, please elaborate the competition effect between surface roughness Vs composition for better understanding of the readers.   9.

Reviewer 2 Report

Authors showed that the degradation of PLLA increases the roughness on film surface and the surface wettability of PLLA exhibits crucial dependence on its crystallinity where water contact angle decreases upon alkali hydrolysis in the case of high crystallinity. Discussions are clearly supported by results and results are important for the surface modification of PLLA. I recommend publication of manuscript in Polymers after minor revision.

"Consequently, the hydrophilicity has been improved significantly." sentence in the abstract should be clarified by explaining for which sample this sentence is valid: high crystalline or low crystalline. This revision is also valid for conclusion. "This is the reason for high-density carboxyl and hydroxyl groups and higher magnitudes of surface roughness." sentence should be clearly include "for the high crystalline case" part.

The preparation of two samples with different crytallinities should be clarified. "PLLA films were prepared by hot-pressing at 200 °C and 10 MPa. The melted PLLA films (with the thickness of 500 μm) were quickly cooled to 120 °C under 10 MPa and annealed for 0 min and 60 min, which were named as PLLA0 and PLLA60, respectively." This sentence can be given in two different sentences to clarify preparation of low and high crystalline samples.

Introduction is slightly weak. Previous results on the hydrolysis of PLLA can be summarized in a paragraph such as:

https://www.sciencedirect.com/science/article/pii/S1364032117307876

https://link.springer.com/chapter/10.1007/12_2016_12

https://www.mdpi.com/1420-3049/26/24/7554

Although this is not the main point of the manuscript, comparison study was performed at fixed temperature and pH. Authors can discuss the effect of temperature and pH on their conclusion: competition effect between surface roughness and composition. "How different physical conditions can effect on this competition" can be discussed shortly.

Round 2

Reviewer 1 Report

Thank you for your patience and accepting all suggestion,

The manuscript can be accepted for publications after minor revisions.